# miR-125-3p and miR-276b-3p Regulate the Spermatogenesis of *Bactrocera dorsalis* by Targeting the *orb2 Gene*

**DOI:** 10.3390/genes13101861

**Published:** 2022-10-15

**Authors:** Summar Sohail, Kaleem Tariq, Muhammad Sajid, Muhammad Waqar Ali, Wei Peng, Hongyu Zhang

**Affiliations:** 1Key Laboratory of Horticultural Plant Biology (MOE), China-Australia Joint Research Centre for Horticultural and Urban Pests, Institute of Urban and Horticultural Entomology, College of Plant Science and Technology, Huazhong Agricultural University, Wuhan 430070, China; 2Department of Environmental Sciences, Kohsar University Murree, Murree 47150, Pakistan; 3Department of Entomology, Abdul Wali Khan University, Mardan 23200, Pakistan; 4Department of Biotechnology, University of Okara, Okara 56300, Pakistan; 5Hunan Provincial Key Laboratory of Animal Intestinal Function and Regulation, State Key Laboratory of Animal Intestinal Ecology and Health, Hunan Normal University, Changsha 410081, China

**Keywords:** *Bactrocera dorsalis*, orb2, miR-125-3p, miR-276b-3p

## Abstract

*Bactrocera dorsalis* is considered a major threat to horticultural crops. It has evolved resistance against insecticides. It is believed that development of new methods is highly desirable to control this destructive agricultural pest. Sterile insect technique is emerging as a potential tool to control this insect pest by reducing their reproductive ability. Here we report that *orb2* has high expression in the testis of *B. dorsalis* which is the target of miR-125-3p and miR-276b-3p and plays a critical role in the spermatogenesis. Dual luciferase reporter assay using HEKT293 cells demonstrates that *orb2* gene is downregulated by miR-125-3p and miR-276b-3p and is a common target of these miRNAs. Dietary treatment of adult male flies separately and in combination of agomir-125-3p (Ago-125-3p) and agomir-276b-3p (Ago-276b-3p) significantly downregulated the mRNA of *orb2*. The combined treatments of agomirs suppressed the level of mRNA of *orb2* significantly more than any single treatment. Altered expression of miR-125-3p and miR-276b-3p significantly decreased the total and live spermatozoa in the testis which ultimately caused reduction in male fertility. Furthermore, we demonstrate that miR-125-3p, miR-276b-3p, and *orb2* dsRNA are the novel agents that could be used in a genetic-based sterile insect technique (SIT) to control the *B. dorsalis*.

## 1. Introduction

*Bactrocera dorsalis* (Hendel) (Diptera: Tephritidae) is one of the most damaging insect pests of fruit and vegetables [1]. *B. dorsalis* is found throughout Asia and has attacked more than 270 host plant species [2,3]. To control its population, indiscriminate use of chemical pesticides has produced high resistance to chemical insecticides and gravely threatened the use of these compounds for effective and sustainable management [1]. For this reason, more effective and new strategies are urgently desired to control this insect pest [4]. Genetic control of the insects provides valuable addition to the armory of already available insect control methods.

The sterile insect technique (SIT) is a genetic control method that relies on the introduction of sexually transmitted factors that reduce the reproductive capacity of the males. In SIT, a large number of sterile males are introduced in the field to compete with males for mating with females, which ultimately lay sterile eggs [5]. Genes related to the spermatogenesis have been proposed as the potential targets to develop any new genetic control strategy [6,7]. Moreover, disruption of spermatogenesis-related genes can cause sterility in males [8].

Spermatogenesis is a multistep and highly orchestrated process which contains three distinct phases, mitosis, meiosis, and spermiogenesis, through which diploid spermatogonia multiply and differentiate into mature spermatozoa [9]. All phases of spermatogenesis are regulated by specific genes which are controlled by non-coding RNAs [10]. The microRNAs (miRNAs) are endogenous non-coding (~22 nt) small RNA molecules that regulate gene expression by binding to the 3′-UTR of target mRNA [11]. Currently, miRNAs research has been revolutionized in agriculture due to diverse roles of miRNAs in biological processes of insect metamorphosis, immunity, metabolism, longevity, and reproduction [12,13,14,15,16,17]. Accumulated evidence reveals that miRNAs regulate genes at different stages of spermatogenesis such as mitosis, proliferating spermatogonia, spermatocytes, and spermiogenesis [18,19,20]. Previous studies demonstrate that certain miRNAs exert synergistic effects in the regulation of different biological processes, for instance, combination of both miR-499 and miR-133 regulate the cardiac differentiation synergistically [21]. Similarly, miR-26a and miR-30c jointly regulate TGFβ1 to induce epithelial-to-mesenchymal transition in diabetic nephropathy [22]. Combined influence of another miR-143 and miR-145 in the breast cancer suppresses the ERBB3 [23], miR-34, and miR-497 target cyclin E1 cooperatively in lung cancer [24]. A large number of miRNAs expressed in the testis of *B. dorsalis* have been identified through deep sequencing [25].

Although *orb2* plays multiple roles in spermatogenesis of Drosophila [26], the exact role of miRNAs to control this gene remains unclear. In the present study, we identified that *orb2* mRNA levels are highly expressed in the testis in comparison to other body tissues of *B. dorsalis*. We hypothesized that the elevated levels of *orb2* were a result of miRNAs regulation. Subsequently, it was identified that miR-125-3p and miR-276b-3p directly target the *orb2* and regulate its expression in the testis of *B. dorsalis*. Furthermore, it was observed that combination of Ago-125-3p and Ago-276b-3p caused a very robust impact on the inhibition of *orb2* mRNA than influence of individual Ago-125-3p and Ago-276b-3p. Subsequently, combined impact of Ago-125-3p and Ago-276b-3p greatly reduced reproductive capacity of *B. dorsalis* compared to separately used miRNAs. To further validate our results, it was investigated that this suppression was due to the inhibition of *orb2* mRNA. To further verify the influence of target miRNAs on the reproductive capacity of males of *B. dorsalis*, RNAi of antagomir-125-3p (Ant-125-3p) and antagomir-276b-3p (Ant-276b-3p) treatment of insects with *orb2* specific dsRNA was performed. As expected, the fertility rate of these insects was recovered which was reduced with *dsorb2*, Ago-125-3p, and Ago-276b-3p treatment.

## 2. Materials and Methods

### 2.1. Rearing of Insects

The adults and larvae of the *B. dorsalis* were reared at 27 ± 1 °C and humidity 75 ± 5% in the laboratory under a 12 h light: 12 h dark photoperiod. Adults were fed with a supplemental diet containing a mixture of 7.5% sugar, 2.5% yeast extract, and 90% water. Banana pulp was used to rear the larvae.

### 2.2. Selection of Target Gene

*Orb2* gene was chosen on the basis of previous studies. Primers for 3′UTR cloning and dsRNA synthesis were designed by using the database (https://www.ncbi.Nlm.nih.gov/ accessed on 23 May 2016) (Appendix A). The procedure has been described [27].

### 2.3. miRNA Target Site Prediction

Target site of miRNAs was predicted by two different bioinformation tools, RNAhybrid [28] and TargetScan [29].

### 2.4. dsRNA, Agomir, and Antagomir Preparation

A total of 500 *orb2* and 495 bp of GFP fragments were amplified using PCR primers and sequenced. Then, T7 promoter sequence was added with PCR primers (Appendix A) and the PCR product was amplified from 1μg cDNA, which was used as template for dsRNA synthesis. dsRNA was synthesized from purified PCR products using kit (T7 Ribo MAXTM Express RNAi System, Promega, Madison, WI, USA). Bands of the expected sizes confirmed the integrity of dsRNA and concentrations of dsRNAs were measured by spectrophotometry (Nano Drop 1000, Thermo Scientific, Ann Arbor, MI, USA). dsRNAs were stored at −80 °C. Agomirs and antagomirs were bought from company Gene Pharma (Shanghai, China).

### 2.5. Dual Luciferase Reporter Assay

The psiCheck-2 reporter was constructed by inserting 3′UTR (1200 bp) fragment harboring the target sites for miR-9c-3p, miR-125-3p, miR-210-5p, miR-276b-3p, and miR-3931-3p into the psiCheck-2 vector. For transfection, HEKT293 cells were cultured in DMEM with 10% FBS. After growing the cells in plates (96-well) with serum-containing medium for 12 h, they were transfected with 100 nM miRNA mimics or negative control (NC) for single mimic, for combination of 125/276b-3p half dose (50 nM for each miRNA mimic), and 100 ng of psiCheck-2 vector containing the 3′UTR of *orb2* gene per well using 0.3 μL fugene HD (Promega, USA). After 48 h of transfection, cell lysis was performed using 1X passive lysis buffer (Promega, USA) and renilla luciferase activities were determined in a dual luciferase assay system (Promega, USA) following the manufacturer’s protocol. Values of firefly luciferase were normalized to renilla and the ratio of firefly to renilla was calculated.

### 2.6. RNA Isolation and Quantitative Real-Time PCR

Total RNAs from *B. dorsalis* testis and other body tissues were isolated using Triazole reagent (Invitrogen, Waltham, MA, USA). For *orb2*, cDNA was synthesized using the Prime Script RT-PCR Kit (TaKaRa, Japan) and quantitative real time (RT- PCR) PCR was performed using the SYBR Premix Ex Taq (TaKara). *B-Actin* was used as an internal control for normalization. Primers for qRT-PCR are given in Appendix A for miR-125-3p, miR-276b-3p, and U6 snRNA, cDNA was synthesized using Stem loop PCR primers. SYBR Green Master Mix (miScript SYBR Green PCR Kit, Qiagen, Hilden, Germany) was used to perform real-time PCR. All quantitative real-time PCR was done on an Applied Biosystems, Carlsbad, CA, USA. The stem loop primers, RT-qPCR primers for miRNA-125-3p, miR-276b-3p, and U6 snRNA were prepared from Ribobio (Guangzhou, China). The 2^−^^△△Ct^ method was used to normalize the expression of *orb2* and miRNAs.

### 2.7. Administration of dsRNA, Agomirs, and Antagomirs to the Adult Flies

dsRNAs, agomirs, and antagomirs were administered to the adult flies through feeding with the same method as previously described by Li et al. [30]. Newly enclosed females and males were placed into separate rearing cages (17 cm × 9 cm × 9 cm). A total of 100 males or females were used for each treatment. Male and female flies were kept starved for 24 h before feeding of dsRNAs, agomirs, and antagomirs. Then, 1 mL agomir (100 nM/µL) and 1 mL dsRNA (1 µg/µL) were added on the plates of artificial diet. In addition, negative control and dsRNA of green fluorescent protein (dsegfp) were used as the controls. After feeding of dsRNA and agomirs for 6 h, all insects were shifted to a normal diet. After one day, insects were collected for 5 consecutive days for qRT-PCR. Three independent biological replicates were performed. One-way ANOVA was used to analyze the results (*p*  <  0.0001, Tukey test). For the rescue experiment, first, we starved the insects for 24 h before antagomirs treatment. After 6 h feeding of antagomirs, insects were shifted to a diet containing dsRNA at the concentration of 1000 ng/μL. After 12 h feeding of dsRNA, insects were again shifted to a normal diet. After 24 h, insects were collected for qRT-PCR analysis for five consecutive days.

### 2.8. Reproductive Capacity of Male Flies

A total of 20 pairs of males (virgin) treated with dsRNA, Ago-125-3p, Ago-276b-3p, and Ago-125-3p/Ago-276b-3p and untreated females (virgin) were placed in a cage for crossing. After mating for 24 h, within 25 min of egg laying, eggs were collected. The eggs were collected with a soft brush on black paper (A4) very carefully and counted. After counting, eggs were placed for hatching in plastic bowls containing mashed banana under controlled laboratory conditions. After 4–5 days, hatched larvae were calculated, and the percentage of reproductive capacity was calculated using online calculator as compared to control.

### 2.9. Sperm Viability Assays and Spermatozoa Counts

The testes of adult males of *B. dorsalis* were dissected in Hayes solution. Testes were pierced with forceps and 2 μL of liquid containing sperms was obtained. This liquid was diluted with 300 μL of Hayes solution and the sperm viability was calculated following the previously established method [31,32]. A sperm viability kit was used to separate dead and live sperms. A total of 5 μL of sperms were incubated with 5 μL of SYBR-14 solution following the previously described method. Sperm viability was measured using microscope (UMNG2, Olympus, Japan). Then, 400 dead and live sperms per slide were calculated at 400× magnifications of microscope. Dual-stained sperms were excluded from the data. Sperm viability for each sample was measured by calculating the percentage of live sperm. Experimental procedures were verified by placing the sample in refrigerator (at −80 °C) for 8 h, sperms were killed. When sperm viability was calculated, all dead sperms observed were stained red. To count the sperm numbers, sperms were fixed on slides in ethanol, dried in air, and then stained with DAPI. After 15 min, spermatozoa were calculated by using a fluorescence microscope.

### 2.10. Data Analyses

Expression of *orb2* and miRNAs was analyzed using LSD with SPSS 2O0 software (SPSS Inc., Chicago, IL, USA). Prism graph pad was used to analyze the effect of agomirs, antagomirs, and dsRNA treatments on miRNA expression, gene expression, male fertility, and total number of spermatozoa. For qRT-PCR and the luciferase assay, a Tukey test was used (*p* < 0.05) and results are represented as the mean  ±  SEM. Reproductive capacity was calculated from the percentage of larvae per eggs and obtained data were analyzed by *t*-test. One-way analysis of variance (ANOVA) was used to analyze all the results.

## 3. Results

### 3.1. Gene Selection

*Orb2* (XM_019991701.1) was chosen based on previous studies as its homologue (NP_648266) plays significant roles in the spermatogenesis of Drosophila [26]. The *orb2* gene of *B. dorsalis* shares 61% identity with its homologue.

### 3.2. Prediction of miRNAs Targeting orb2 and Their Confirmation

miRNAs prediction through RNA hybrid showed that the 3′UTR of *orb2* consists of binding sites for 5 miRNAs, miR-9c-3p, miR-125-3p, miR-210-5p, miR-276b-3p, and miR-3931-3p. Moreover, another prediction software, for instance, TargetScan, was used to further predict our findings about binding sites (Figure 1). To further investigate the relation between *orb2* and miR-9c-3p, miR-125-3p, miR-210-5p, miR-276b-3p, and miR-3931-3p in vitro, the fragment (1200 bp) of *orb2* 3′UTR containing the nucleotides complementary to these miRNAs was cloned into a luciferase reporter plasmid (psicheck-2vector) by integrating the *orb2* 3′UTR downstream of the renilla luciferase reporter gene of psicheck-2 vector using NotI and XhoI sites. Cells were co-transfected with *orb2* 3′UTR containing plasmid and mimics of miR-9c-3p, miR-125-3p, miR-210-5p, miR-276b-3p, and miR-3910-3p separately. Mimics of miR-125-3p and miR-276b-3p significantly reduced the luciferase signals 57% and 53% (Figure 1B). Reduction in luciferase signals suggests that these miRNAs might have a relation with *orb2*. Comparatively, miR-9c-3p, miR-210-5p, miR-3931-3p, and negative control exhibited no change in the luciferase signals (Figure 1B).

### 3.3. Expression Profiles of orb2 and Its Target miRNAs

To confirm the relation between *orb2* and its target miRNAs, the tissue distribution of *orb2* mRNA (Figure 2A), miR-125-3p, and miR-276b-3p was determined by means of stem-loop quantitative reverse transcriptase-polymerase chain reaction (qRT-PCR) (Figure 2B,C). The overall expression of miR-125-3p, miR-276b-3p and their target gene (*orb2*) exhibited opposite patterns in the adult testis of *B. dorsalis*. More specifically, *orb2* was highly expressed in testes. In contrast, the expression of miRNA-125-3p and miR-276b-3p was low in testes. Real-time PCR performed 24 h administration through feeding for 6 h. Successful delivery of agomirs and antagomirs of miR-125-3p and miR-276b-3p was demonstrated by increase and decrease in the level of miR-125-3p and miR-276b-3p (Appendix A). With the feeding of Ant-125-3p, the level of *orb2* mRNA was increased in the 1st, 2nd, 3rd, and 4th day 1.74-fold, 2.29-fold, 1.81-fold, and 1.45-fold, respectively, as compared to control, whereas a non-significant difference was observed in the 5th day as compared to control (Figure 3A).

### 3.4. Overexpression of miR-125-3p and miR-276b-3p Impairs the Male Fertility

To study the impact of *orb2* mRNA inhibition and over-expression of miR-125-3p, miR-276b-3p on male fertility of *B. dorsalis*, the newly emerged males were treated with *dsorb2* and agomirs of miR-125-3p and miR-276b-3p. When males became 12 days old, the treated males were crossed with virgin untreated 12 days old wild type females. Then, various effects on male fertility were investigated. In *dsorb2*, Ago-125-3p, and Ago-276b-3p treated individuals, sterility was noticed 71.19%, 67%, and 69%, respectively, as compared to their controls (Figure 4A–C). These results suggested that *dsorb2*, Ago-125-3p, and Ago-276b-3p have a remarkable impact on male fertility (Figure 4A–C). In addition, combination of Ago-125/276b-3p exhibited greater effects on male fertility than any single miRNA. After feeding the combination of Ago-125/276b-3p orally, egg hatching percentage was calculated. As expected, hatching percentage was reduced to a greater extent. Only 17% hatching was found as compared with the control group (Figure 4D).

### 3.5. Investigation of Number of Sperms and Sperm Viability

The reduced hatching rate prompted us to find the reason of impaired fertility. We conducted a bioassay to quantify the total number of sperms in the testes of treated males with Ago-125/276b-3p combination and in control males. As expected, very significant reduction was found in the average number of sperms in Ago-125/276b-3p combination treated male flies (Figure 5A) as compared to control flies. It is currently not confirmed whether decrease in the number of sperms is the main reason for sterility in males (83%). We assumed that there are multiple reasons for reduced fertility. Thus, the viability of sperms in the treated and control flies were examined. Interestingly, more dead sperms were found than live sperms in the Ago-125/276b-3p treated males than control males. Total number of sperms and live sperms in treated flies was reduced up to 57% and 52%, respectively, as compared to the control group (*p* < 0.05) (Figure 5A–C).

**Figure 3 genes-13-01861-f003:**
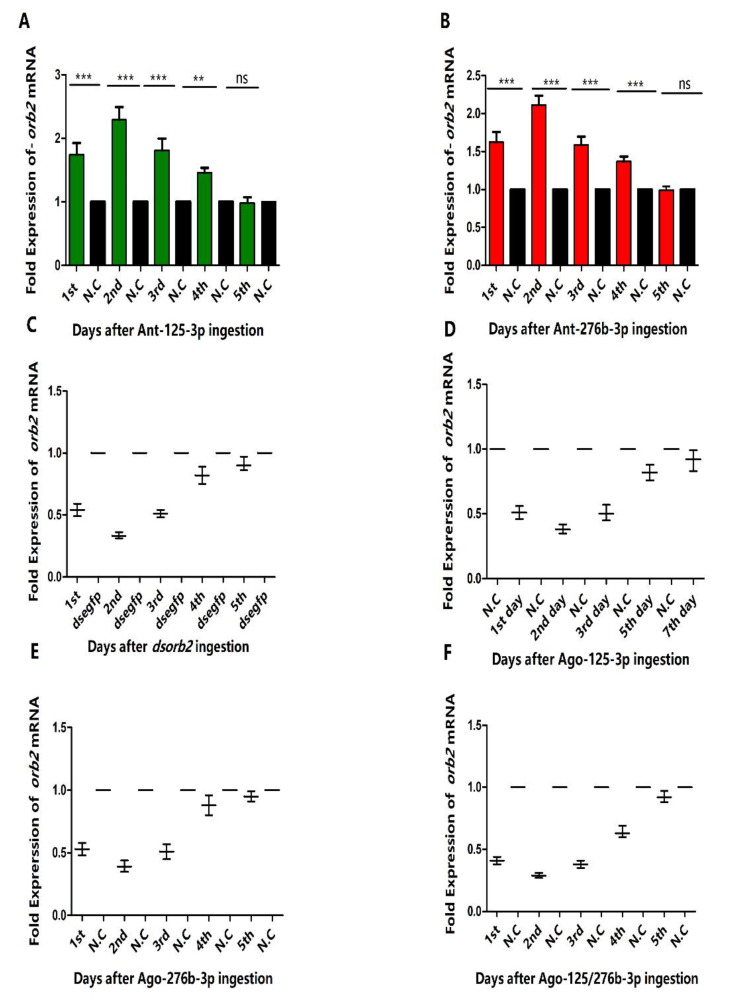
Effects of *dsorb2*, agomirs, and antagomirs of miR-125-3p, miR-276b-3p on mRNA of *orb2.* (**A**) Fold expression of *orb2 mRNA* after ingestion of antagomir-125-3p. (**B**) Fold expression of *orb2 mRNA* after ingestion of antagomir-276b-3p. (**C**) Fold expression of *orb2* mRNA after ingestion of *dsorb2*. (**D**) Fold expression of *orb2* mRNA after ingestion of agomir-125-3p. (**E**) Fold expression of *orb2* mRNA after ingestion of agomir-276b-3p. (**F**) Fold expression of *orb2* mRNA after ingestion of agomir-125/276b-3p combination. miR-125-3p and miR-276b-3p expression was normalized to U6 and *orb2* mRNA was normalized to β actin. ”N.C” represents negative control.

**Figure 4 genes-13-01861-f004:**
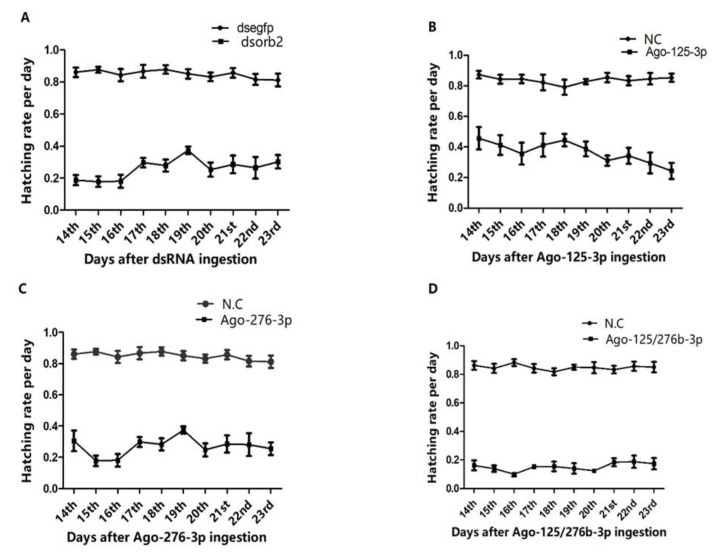
*orb2*, miR-125-3p, and miR-276b-3p are required for *B. dorsalis* male fertility function. (**A**) Egg hatching rate in response to *dsorb2*. (**B**) Egg hatching rate in response to agomir-125-3p. (**C**) Egg hatching rate in response to agomir-276b-3p. (**D**) Egg hatching rate in response to agomir-125/276b-3p combination as compared to controls.

### 3.6. RNAi of Antagomir-125-3p and Antagomir-276b-3p Treated Individuals Rescued the Phenotype Caused by dsorb2 Treatment

To further verify that *orb2* is the authentic target of miR-125-3p and miR-276b-3p in the testis, we conducted the RNA interference (RNAi) experiments of Ant-125-3p and Ant-276b-3p treated males of *B. dorsalis*. Adverse phenotypes caused by depletion of *orb2* with dsRNA, Ago-125-3p, and Ago-276b-3p were expected to be recovered. In our rescue experiment, when we did the RNAi of Ant-125-3p and Ant-276b-3p treated individuals, the expression level of *orb2* mRNA was reached at the same level as in the dsegfp group (Figure 6A,B). Furthermore, we were also inspired to investigate whether it has any effect on reproductive capacity of males of *B. dorsalis*. Therefore, we performed the egg laying and hatching assays of Ant-125-3p/*dsorb2*, Ant-276b-3p/*dsorb2*, and *dsegfp* (control). As expected, no significant difference was found in the egg hatching rate as compared to the control group (Figure 6C,D). These results indicated that the reproductive capacity reduced by feeding *dsorb2*, Ago-125-3p, and Ago-276b-3p was only due to the depletion of *orb2* mRNA.

**Figure 5 genes-13-01861-f005:**
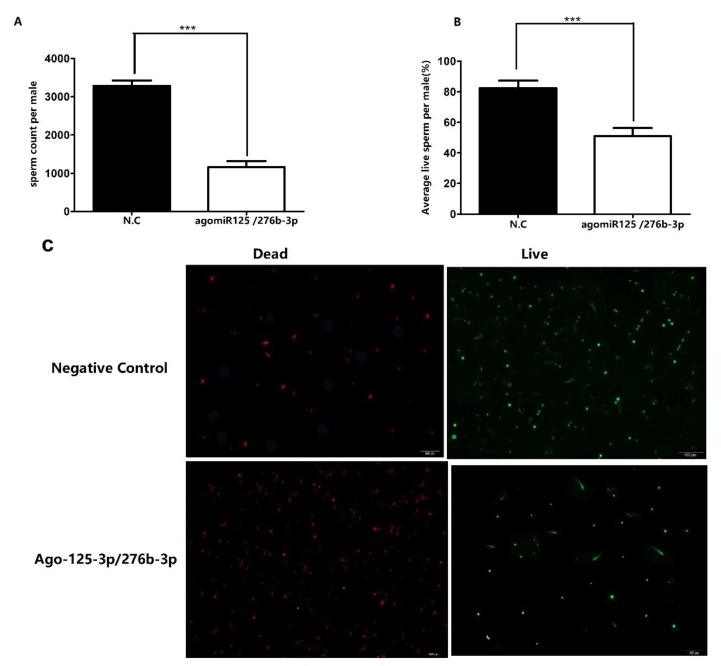
Reduced number of total sperms and live sperms in the *orb2* knockdown males. (**A**) Total number of sperms significantly decreased in flies after feeding with combination of agomir-125-3p/276b-3p as compared with flies feeding negative control agomirs. (**B**) Percentage of live sperm significantly decreased in the testis of flies feeding with combination of Ago-125/276b-3p as compared with flies feeding with negative control agomirs. (**C**) represents live and dead sperms after ingestion of agomir-125/276-b-3p combination. Red color indicates dead sperms, green color indicates live sperms.

## 4. Discussions

In recent years, the sterile insect technique has gained more attention to control insect pests [33,34]. The ideal ambition of this technique is to find the substitute of chemical control. To contribute to pest control, this new environmentally friendly technique should be fully developed. In the start, irradiation-based SIT was used to produce sterile males. Initially, it was envisioned that this technique might have controlled target pests in the field. However, previous studies have demonstrated that use of this technique is not straightforward [35]. On the contrary, other investigators have shown that SIT can control the pests [36]. Now, many scientists are using the genetic methods instead of irradiation to make the insects sterile [37]. Therefore, new tools to develop more efficient sterile insect techniques is highly desirable.

In the present study, we have demonstrated that Ago-125-3p and Ago-276b-3p can be used effectively in silencing the orb2 gene to control the insect pests (Figure 3A–C). The miRNAs are small non-coding RNAs that function mainly post-transcriptionally by affecting the stability or translation of their target mRNAs [38,39]. A single miRNA can potentially regulate hundreds of genes [40]. We have shown that two miRNAs target a single gene in regulating the spermatogenesis of *B. dorsalis*. We selected the *orb2* gene which is related to spermatogenesis of *B. dorsalis* on the basis of previous studies [40]. First, we predicted five miRNAs which can target this gene by using the bioinformatics tool RNAhybrid (Figure 1A). Afterwards, the dual luciferase reporter assay was performed by using a well-established cell line for miRNAs that mimics transfection. Our results clearly showed that miR-125-3p and miR-276b-3p suppressed the luciferase activity significantly (Figure 1B).

**Figure 6 genes-13-01861-f006:**
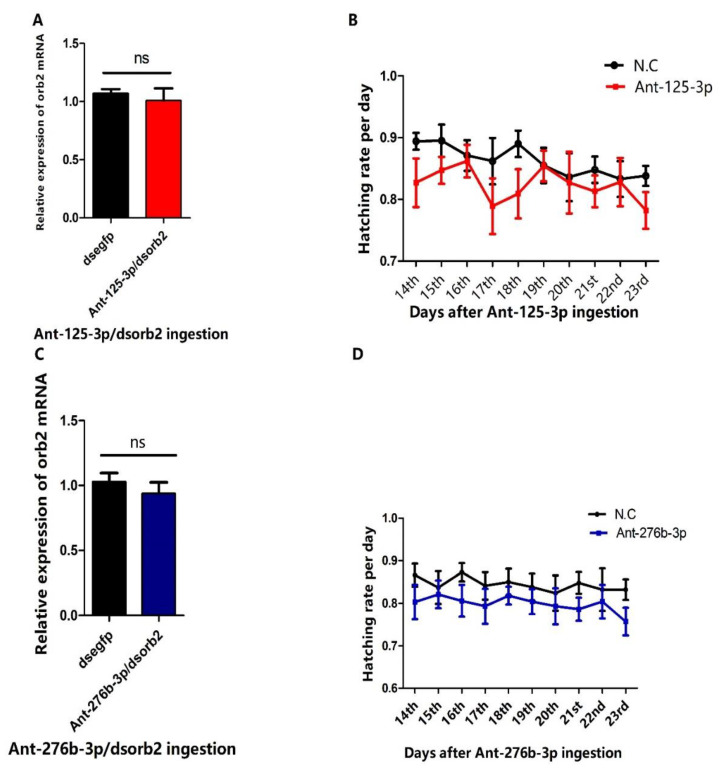
RNAi of antagomir treated individuals rescued the phenotype. (**A**) RNAi of Ant-125-3p treated individuals. (**B**) RNAi of Ant-276b-3p treated individuals. (**C**) Effects of RNAi of Ant-125-3p treated individuals on the reproductive capacity of males of *B. dorsalis*. (**D**) Effects of RNAi of Ant-276b-3p treated individuals on the reproductive capacity of males of *B. dorsalis*.

Currently, it is unclear whether combination of these two miRNAs would result in a synergistic action. To test our hypothesis, the dual luciferase reporter assay was performed by using combination of miR-125/276b-3p mimics for transfection. Interestingly, when co-transfection was carried out with miR-125/276b-3p combination, the results were found remarkably superior compared with the miR-125-3p and miR-276b-3p alone (Figure 1C). Thus, miR-125-3p and miR-276b-3p were selected for further study due to their high impact on luciferase activity.

Furthermore, the expression of *orb2* was first analyzed in different body tissues. High expression of *orb2* in the testis of *B. dorsalis* males suggests that this gene may be important for the spermatogenesis of *B. dorsalis* (Figure 2A). In contrast, expression of *orb2* in drosophila is high in the central nervous system. The expression difference may be caused by the specificity of *orb2* genes in different species of insects. Next, the expression of miR-125-3p and miR-276b-3p was also measured in different tissues of males of *B. dorsalis*. As expected, their expression was lower in testes than *orb2* (Figure 2B,C). Moreover, oral administration of Ago-125-3p and Ago-276b-3p suppressed the mRNA expression of *orb2* (Figure 3C,D). The suppression of mRNA of *orb2* resulted in decreased hatching rate, sperm quantity, and viability. This phenotype resembled the hypomorph alleles of *Drosophila melanogaster orb2* gene in which the partial sterility is exhibited in contrast to the null alleles which resulted in complete sterility [26]. This explains the fact that miRNA-125-3p, miR-276b-3p, and RNAi of *orb2* failed to completely suppress the *orb2* mRNA.

In Drosophila, mutants of *orb2* exhibited blocked meiosis and 16-cell cysts duplicated their DNA. At this stage, these cysts attempted to differentiate into mature sperm but failed [41]. The individualization process is accomplished by individualization complex. In the mutants of *orb2* in Drosophila, IC were never assembled in the testis and the individualization was disrupted. Once individualization was compromised, no mature sperms were found in the testis.

miRNA inhibitors are used to study their functional role, which is based on nucleic acid molecules that suppress miRNA expression and function. In miRNA synthesis, reverse complement of the mature miRNA incorporated and chemically modified to avoid RISC-induced cleavage make them resistant to nucleolytic degradation and increased binding affinity [42]. When applied in vivo, the binding of endogenous miRNAs to chemically prepared target sites is believed to be irreversible, and assumed to sequester the endogenous miRNA, restricting it to perform normal function [43,44]. To get the expected outcomes of miRNA inhibitors, it is crucial to understand the degree to which a miRNA inhibitor synthesized against one miRNA interferes with other miRNAs. The miRNA seed region (position 3 to 8) and a 3′ region (position 13 to 18) is very important for the recognition of the miRNA inhibitor targeted by endogenous miRNAs for the type that is chemically modified [45]. In the rescue experiment, the RNAi of Ant-125-3p and Ant-276-3p treated individuals was performed (Figure 6C,D). As previously conducted in mosquitos [17], fertility was recovered. This demonstrates that *orb2* is a direct target of these two miRNAs.

## 5. Conclusions

In testes, it was found for first time that *orb2* could regulate sperm production. Meanwhile, it was verified that *orb2* was the target gene of two candidate miRNAs, miR-125-3p and miR-276b-3p. Moreover, the current study showed that the combination of miR-125-3p and miR276b-3p is the most effective for producing sterile males. These findings on the function of miRNAs in spermatogenesis would provide an important basis for further elucidation of miRNAs in regulating the spermatogenesis of *B. dorsalis*.

In light of our experimental findings, including the spermatogenesis suppressor activities of miR-125-3p and miR-276b-3p, both miRNAs could be promising candidates to improve genetic sterile insect techniques. These results could also be potentially utilized in the development of new genetic insect control technique. However, the specific molecular mechanism on sperm development and maturation in insects still needs further exploration.

## Figures and Tables

**Figure 1 genes-13-01861-f001:**
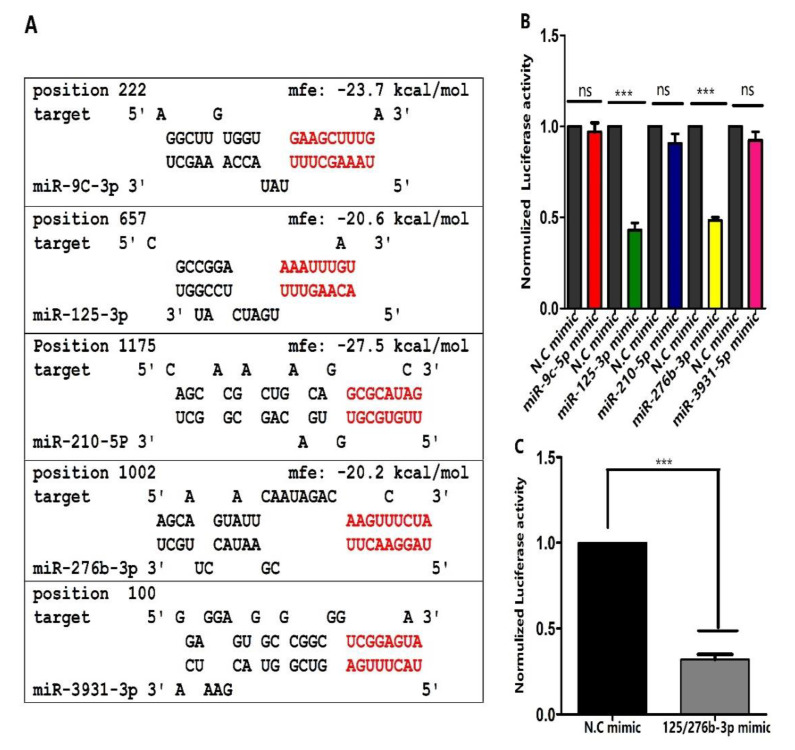
miRNAs targeting *orb2* as predicted by RNAhybrid and confirmation through dual luciferase reporter assay. (**A**) Potential miRNA target sites of miR-9c-3p, miR-125-3p, miR-210-5p, miR-276b-3p, and miR-3931-3p. Seed sequence of the miRNAs and their putative binding sites in the 3′UTR are indicated by red letters. mfe: indicate match free energy value. (**B**) Effects of miR-9c-3p, miR-125-3p, miR-210-5p, miR-276b-3p, and miR-3931-3p on luciferase activity. (**C**) The effects of combination of miR-125-3p/276b-3p on luciferase activity. Treatments were compared with their respective controls using ANOVA (Tukey-test, *p* < 0.05). *** indicates *p* < 0.001 and ns non-significant respectively.

**Figure 2 genes-13-01861-f002:**
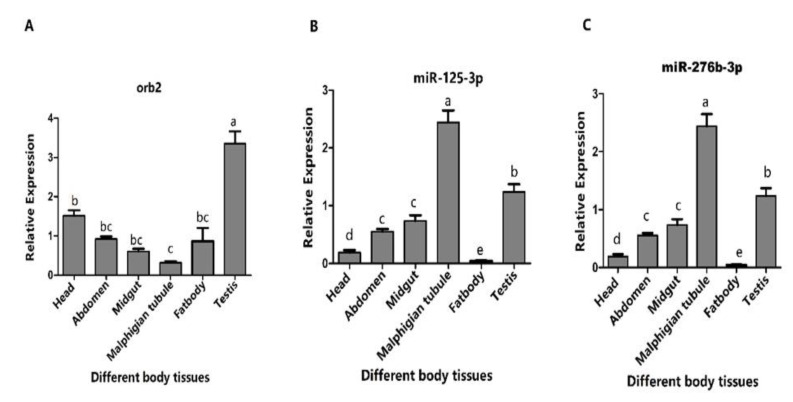
Expression profiles of *orb2* and its target miRNAs in different tissues of adult males of *B. dorsalis.* (**A**) Relative expression of *orb2* in different tissues. (**B**) Relative expression of miR-125-3p in different tissues. (**C**) Relative expression of miR-276b-3p in different tissues. Different letters above the bars indicate significant differences (least significant difference in one-way analysis of variance (*p* < 0.05).

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
