# Peer review of "miR-125-3p and miR-276b-3p Regulate the Spermatogenesis of *Bactrocera dorsalis* by Targeting the *orb2 Gene"

_genes, 2022, doi:10.3390/genes13101861_

Round 1
Reviewer 1 Report
Comments,
Overall, the ms needs to be reorganized/revised for proper presenting of results.
Fig. 1 panel A, please change MiR-3931-3p to miR-3931-3p
Line 97-98, did authors sequenced the PCR product of orb2 to confirm its accuracy?
Line 175-176, please clarify this part, how did you calculated the number of eggs laid per day and rate of eggs hatching by t-test ? T-test is a method to compare the significance of treatment and control.
Line 181, is 61% identity at AA or nucleotide level ?
Line 145, you said ‘20 pairs of each treated males (virgin) and untreated females (virgin) were placed…’, please clarify here treated with which molecules, dsorb2 ? antagomirs or combined ?
Line 199-200, here please clarify if you contransfected HEK293 cells with five miRNAs combined or separately
Line 204, you said ‘3.3 Expression profiles of orb2 and its target miRNAs’, please explain how miRNAs become target of orb2 ?
Line 210-211, the authors claimed that the miRNA-125-3p and miR-276b-3p were highly expressed in other tissues as compared to testis, but I don’t see its high expression in fatbody and head.
Fig. 3 panel A and B, the y-axis title is fold change of orb2 gene, but in Fig. 3 legend, it says mi125-3p and mi276b-3. Also miR-276b-3 is incorrect, it should be miR-276b-3p, unless authors used different miR molecule in experiments. In panel D, data on 7th day was presented, no 4th day (?)
Line 238-239 Fig. 3 panel E, what is the combination ?
Line 241-242 the authors said ‘miR-125-3p and miR-276b-3p expression was normalized to U6, but I don’t see any expression analysis in Fig. 3.
Line 243-244 repeatedly showed on line 255-257
Line 245, the ‘3.5. Overexpression of miR-125-3p and miR-276b-3p impairs the male fertility’ showed here again, authors should check/read the ms carefully before submitted for review.
Line 292-296 if you treated male flies with both ant-125-3p/dsorb2, then the expression of orb2 gene should be suppressed by dsorb2, why the expression level of orb2 is same as that in control?
Line 343-344 this suggestion here is not proper, there many other reasons associated with sperm quality, unless there are reports talking about the involvement of orb2 gene in sperm polarization.
Reviewer 2 Report
The Manuscript [ijerph-1920669] entitled (miR-125-3p and miR-276b-3p regulate the spermatogenesis of 2Bactrocera dorsalis by targeting the orb2 gene) confirmed that orb2 has high 19expression in the testis of B. dorsalis which is the target of miR-125-3p and miR-276b-3p and play 20critical role in the spermatogenesis. These findings will be useful in sterile insect technique (SIT)
In general, All experiments are well designed and explained. The manuscript has good data and results those are introduced and written very well. Here, my comments for the authors that are considered as minor revision:
1- Line 29: SIT, should be in complete words in the abstract. Then, in introduction also use complete words and after abbreviate it.
2- Line 33: mention the family and order of Bactrocera dorsalis (Hendel)
3- Lines 38-39: add reference
4- Line 64: words [more recently] are not suitable for a reference in 2016.
5- Line 87: [diet containing mixture of sugar and yeast extract]. Explain the ratio and volumes.
6- Line 97: don’t start with number [500], write as Five hundred., the same in lines 133, 135, 145.
7- Lines 197-180: it is not suitable in results, use it in methods or discussion.
8- Lines 240-241: [Three independent ….Tukey test)]. Transfer to methods section.
9- Line 287: [as previously conducted in mosquito [32]]. Transfer to discussion section.
10- Lines 301-303[dorsalis. One-way analysis …. at P < 0.05]. remove from here and explain that in sub-section of statistical analysis.
11- Lines 325-327: [Currently, .. for transfection.]. Transfer to discussion section
12- Add section of Conclusion[ lines 363-366] and explain more about the results.
13- References: All references need to re-written: Mention all autrhors NOT et. al.,… .Delete [p] before pages. Year should after Journal name. Please check carefully the guidelines for authors.
